# PED-X-Bench: A Benchmark of Adult-to-Pediatric Extrapolation Decisions in FDA Drug Labels

## Abstract

Pediatric clinical trials are ethically complex, expensive, and often infeasible, leading the U.S. FDA to extrapolate adult efficacy and safety data when justified. Yet no public resource systematically documents these regulatory decisions. We introduce **PED-X-Bench**, the first dataset and benchmark that encodes FDA pediatric extrapolation outcomes as a four-way classification task (*Full*, *Partial*, *None*, *Unlabeled*). PED-X-Bench contains 737 drug-label sections (~1M words) spanning 2007–2024 across all therapeutic areas. A two-stage *o3-mini* prompting pipeline mined evidence directly from FDA labels, and nine domain experts adjudicated a stratified sample of 135 records ($\kappa = 0.72$, macro-F1 = 0.63). For each drug, we release the gold-standard extrapolation label, concise efficacy and PK/safety summaries, and harmonized study metadata. We benchmark a diverse suite of baselines from metadata-only classifiers to domain-adapted transformers and show that substantial headroom remains, highlighting the task's complexity. Beyond benchmarking, PED-X-Bench enables the development of AI-assisted regulatory decision-support tools, and safety-focused applications aimed at accelerating pediatric drug development and reducing off-label use. Dataset card, code, and annotations will be released publicly upon acceptance.

## 1 Introduction

Despite decades of effort, off-label medicine use in pediatric populations remains high—about 40 % overall and up to 90 % in neonates (Sachs et al., 2012). Because growth and maturation reshape pharmacokinetics and disease biology, transplanting adult evidence into children can yield subtherapeutic dosing, reduced efficacy, or elevated adverse-event risk (Smits et al., 2022; Tefera et al., 2017; Bellis et al., 2014). Conducting well-controlled pediatric trials is particularly challenging due to small sample sizes, heightened ethical scrutiny, and substantial developmental heterogeneity, all of which increase cost and complexity.

To address these barriers, the Best Pharmaceuticals for Children Act (BPCA) and Pediatric Research Equity Act (PREA) (noa, 2018; 2024) were enacted in the US, requiring and incentivizing pediatric studies. BPCA ("carrot") grants six months of market exclusivity for conducting trials, while PREA ("stick") mandates them for many approvals. These policies have driven over 800 pediatric labeling changes (Wharton et al., 2014), but many approvals still rely heavily on *extrapolation*—using adult efficacy, safety, and pharmacokinetic (PK) data to support pediatric indications.

The 2024 ICH E11A guideline (International Council for Harmonisation (ICH), 2024) introduced a structured, stepwise approach to extrapolation based on disease similarity, drug pharmacology, and treatment response. Crucially, it reframes extrapolation as a continuum rather than a binary decision. Yet this poses computational challenges: the continuum lacks standardization and clear supervision signals, even though machine learning systems require consistent labels to learn predictive patterns. While clinical decisions may operate in gray areas, machine learning systems, especially those trained on textual regulatory data, benefit from structured, consistent labels to learn predictive patterns and support reproducibility. Moreover, in practice, regulatory decisions are often interpreted in coarse categories *full*, *partial*, or *none* as formalized in FDA guidance (Dunne et al., 2011).

Despite its central role in pediatric drug development, no public dataset systematically encodes these extrapolation decisions. As a result, regulatory reasoning remains opaque and inaccessible to computational analysis. We present **PED-X-Bench** to address this gap: a large-scale benchmark that couples a four-way extrapolation taxonomy with LLM-generated rationales across 737 FDA drug labels, enabling the first structured, machine-readable study of extrapolation decisions.

To demonstrate its utility, we benchmark nine models spanning metadata-only, domain-adapted transformers, and fusion approaches, establishing baseline performance for future work in pediatric drug development, clinical NLP, transfer learning, and regulatory science.

Our main contributions are:

- **First systematic extrapolation benchmark:** 737 FDA labels annotated with four-way decision outcomes (Full/Partial/None/Unlabeled), linked to supporting rationales, efficacy and safety summaries, and harmonized metadata.
- **Scalable annotation methodology:** A two-stage LLM pipeline that achieves expert-level performance across three model families, cutting manual labeling effort by 10× while maintaining reliability through a gold set of 135 labels reviewed by nine experts ($\kappa = 0.79$).
- **Comprehensive baseline evaluation:** Nine models spanning metadata-only, domain-adapted transformers, and fusion approaches establish initial performance benchmarks.

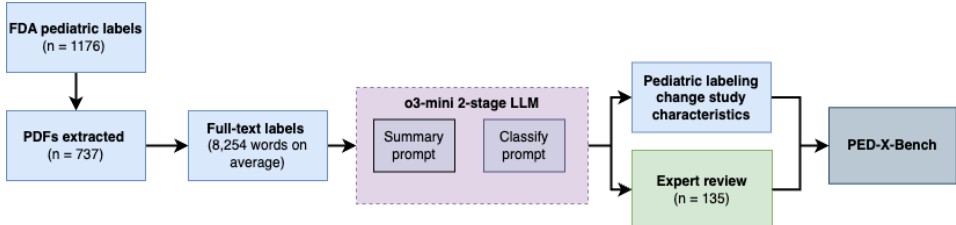

Figure 1: Two-stage LLM pipeline used to build *PED-X-Bench*.

## 2 RELATED WORK

**Pediatric drug development and extrapolation framework.** Since the introduction of BPCA and PREA in the early 2000s, regulatory approaches to pediatric drug development have evolved substantially. Extrapolation has become increasingly common: between 2009–2014, complete, partial, or no extrapolation occurred in 34%, 29%, and 37% of drugs, respectively, shifting to 51%, 23%, and 26% by 2015–2020 Sun et al. (2017); Samuels et al. (2023). This reflects growing confidence in adult-to-pediatric data transfer when disease course and treatment response are sufficiently similar. Yet despite its centrality, these decisions remain undocumented in any public, machine-readable resource, limiting systematic analysis and computational modeling of regulatory reasoning.

**NLP for regulatory documents and biomedical benchmarks.** Most NLP work on FDA labels has targeted safety surveillance rather than regulatory decision-making. Early pipelines like cTAKES Savova et al. (2010) and rule-based systems focused on condition extraction, while newer models such as RxBERT achieved 86.5 F1 on regulatory classification tasks Wu et al. (2023). Other efforts include drug product information extraction for regulatory guidance Shi et al. (2021) and LLM-based approaches to toxicity extraction Silberg et al. (2024), adverse event detection Wu et al. (2025), and label summarization Wu et al. (2024). However, extrapolation decision-making remains computationally unexplored.

In parallel, biomedical NLP benchmarks such as BLUE Peng et al. (2019) and BLURB Gu et al. (2022) have standardized tasks like named-entity recognition, classification, and relation extraction, spurring specialized transformers such as BioBERT Lee et al. (2020), ClinicalBERT Alsentzer et al. (2019), and PubMedBERT Gu et al. (2022). Benchmarks like MedNLI Romanov & Shivade (2018), BioASQ Nentidis et al. (2023), and PubMedQA Jin et al. (2019) have further advanced domain-specific reasoning. Yet none address the regulatory reasoning that underpins drug development decisions.

## 3 THE *PED-X-Bench* DATASET

**PED-X-Bench** is the first openly available corpus that encodes FDA extrapolation decisions as a four-class classification task. The dataset comprises 737 labelling sections issued between 2007 and 2024 in which pediatric indications were considered. For each label we supply three artifact groups: (i) a categorical extrapolation label; Full, Partial, None or Unlabeled, derived from regulatory text; (ii) concise, large-language-model-generated summaries of pediatric efficacy, PK, and safety evidence that have been manually verified for accuracy as well as rationales for its decisions; and (iii) cleaned metadata that capture indication, approval year, brand and generic names, study design, sample size, age range, study centres, participating countries and, where BPCA trials are involved, race and ethnicity information.

**Category definitions** Our categorical definitions mirror regulatory practice. *Full* is assigned when disease progression, treatment response, and exposure–response are sufficiently concordant for pediatric PK (with or without targeted safety data) to substitute for efficacy evidence. *None* denotes clear dissimilarity that requires at least one adequate pediatric efficacy trial. *Partial* occupies the middle ground in which disease and therapeutic response appear similar but uncertainty persists about exposure–response concordance, prompting anything from a single pediatric trial to PK/PD bridging studies to confirm effect. *Unlabeled* applies when no pediatric data are available.

By transforming implicit FDA reasoning into explicit, machine-readable labels and anchoring each to its supporting text and metadata, PED-X-Bench provides a reproducible platform for tasks ranging from classification to evidence retrieval and policy analysis, and it lays a foundation for safer, evidence-based pediatric therapeutics.

Table 1: Extrapolation categories used in *PED-X-Bench*.

| Category | Brief description |
|---|---|
| None | At least one adequate, well-controlled pediatric efficacy RCT conducted; no borrowing from adult data. |
| Partial | Adult efficacy RCT(s) accepted; pediatric PK $\pm$ safety studies bridge exposure–response or confirm dose. |
| Full | pediatric evidence limited to PK (or dosing equations) and safety; adult efficacy is fully borrowed. |
| Unlabeled | No pediatric data reported; extrapolation status not specified in the public label. |

## 4 METHODS

### 4.1 DATA SOURCE AND PREPROCESSING

To create PED- X-Bench, we began with the FDA spreadsheet that lists every pediatric labelling change triggered by BPCA, PREA, or the legacy pediatric Rule since 2007[1]. Each pediatric labeling change includes the date of the pediatric labeling change, specific drug or biological product, indication(s) studied, a summary of the labeling change, therapeutic category and type of legislation. It also contains pediatric study characteristics for the clinical trials conducted to support each pediatric labeling change, including the study number, type of study, study design, number of pediatric participants, ages studied, number of study centers, number of countries and, for BPCA clinical trials, any available racial and ethnic information.

From the most recent release we *(i)* removed veterinary or device entries, *(ii)* collapsed obvious reformulations, and *(iii)* retained the 737 human drugs whose English PDF labels were available. All PDFs were downloaded, converted to plain text with `pdftotext`, and lightly cleaned to preserve page breaks and tables.

---

[1] https://www.fda.gov/science-research/pediatrics/pediatric-labeling-changes

## 4.2 LLM PIPELINE FOR EXTRAPOLATING LABELS

To generate drug extrapolation categories from FDA labels, we devised a two-stage chain-of-thought pipeline using *o3-mini*. A *summary prompt* first scans the full label and extracts every sentence that cites pediatric efficacy, PK, or safety evidence. The resulting evidence block is then condensed into an intermediate summary limited to ≤150 words. The condensed evidence is then passed to a *classification prompt* that *(i)* assigns one of four outcomes (Full, Partial, None, Unlabeled), *(ii)* provides the supporting rationale, and *(iii)* generates final efficacy and safety summaries. Figure 4 illustrates both prompts.

This summary length (≤150 words) was chosen for two practical reasons: (i) it ensures human-in-the-loop evaluability, as prior work shows that annotator agreement and throughput decline once summaries exceed a short paragraph Krishna et al. (2023); and (ii) it aligns with norms in biomedical summarization, where major datasets and journal guidelines adopt comparable lengths (e.g., PubMed and arXiv scientific-paper summaries average 205 and 163 words, respectively).

We performed three **ablation studies**: *(a)* a single-stage classifier over the full label; *(b)* the same two-stage pipeline run with *gpt-4o-mini*; and *(c)* the single-stage prompt plus a verifier that rereads the label and fixes JSON formatting errors.

We also compared performance across three LLM families (GPT o3-mini, Gemini 2.5 Pro, Claude Sonnet 3.7) using identical prompts on 135 expert-annotated labels.

## 4.3 EXPERT ADJUDICATION

For manual review, we recruited nine annotators – eight biomedical data scientists and one clinician – who independently reviewed 135 randomly sampled labels spanning a broad therapeutic mix (antibacterials, antihistamines, antiepileptics, asthma agents, oncology, antivirals). Annotators followed a written guide, recorded an extrapolation label, pasted verbatim efficacy and safety evidence with page references, and flagged uncertainties.

Table 2: Per-class inter-annotator agreement and accuracy on the expert-annotated subset (135 labels).

| Category | $\kappa$ | Accuracy |
|---|---|---|
| None | 0.804 | 0.869 (86/99) |
| Partial | 0.806 | 0.939 (93/99) |
| Full | -0.006 | 0.000 (0/1) |
| Unlabeled | 0.757 | 0.667 (12/18) |
| **Overall** | **0.790** | – |

Each annotation required ~45 minutes of expert time, representing a substantial manual investment. This annotated subset constitutes ~18% of the entire dataset with comparable coverage to prior FDA labeling corpora Silberg et al. (2024); Wu et al. (2025).

Inter-annotator agreement after consensus discussion was high, with an overall Cohen's $\kappa = 0.79$, indicating *substantial agreement* under the Landis and Koch (1977) scale Landis & Koch (1977). Agreement varied by class as shown in Table 2. These 135 gold-standard labels form the test and validation set, while the remaining 602 machine-labeled records provide silver-standard training data.

## 4.4 BASELINE CLASSIFIERS

We benchmark nine complementary models on a fixed train/dev/test split of 687/85/34 labels, spanning metadata-only approaches, domain-adapted transformers, and fusion methods.

**Metadata-based models** We extracted 253-dimensional feature vectors combining z-scored numeric features (patient counts, ages) and one-hot encoded categorical variables (legislation type, therapeutic area). Logistic regression with class-balanced weights and XGBoost were tuned via grid search to optimize macro-F1 on the development set.

**Text-based models** We evaluated generic (BigBird RoBERTa) and domain-adapted transformers (ClinicalModernBERT, BioClinicalBERT) using both full fine-tuning and linear probing. All models employed early stopping on dev loss with consistent preprocessing and optimization.

**Fusion model**  XGBoost fusion combined ClinicalModernBERT [CLS] embeddings with structured metadata, concatenating 768-dimensional text representations with the 253-dimensional feature vectors.

**Evaluation metrics.**  We evaluate model performance using three complementary metrics that account for the multi-class imbalance of pediatric extrapolation. **Accuracy** offers an overall performance snapshot but can overestimate effectiveness given the skewed class distribution (60% None, 31% Partial, 3% Full, 6% Unlabeled). **Macro-F1** averages F1 scores across all classes without weighting by frequency, ensuring that rare but clinically important categories (particularly *Full* extrapolation) contribute equally. **AUC** (one-vs-rest) measures ranking quality independent of classification thresholds, reflecting how well models separate categories.

To quantify uncertainty, we report 95% confidence intervals (CIs) for accuracy and macro-F1 using nonparametric bootstrap resampling (1,000 iterations). Given the rarity of *Full* cases (22 total), macro-F1 is particularly critical, as accuracy alone would over-reward majority-class predictions.

Detailed hyperparameters, training configurations, and computational requirements are provided in Appendix F.4.

## 5  RESULTS

Here we present the details about the PED-X-Bench dataset, then we discuss our validation with human annotated dataset and the effect of ablations on performance. Finally we illustrate the utility of our dataset as a benchmark in data extrapolation classification tasks.

### 5.1  DATASET OVERVIEW

Figure 2 shows that *None* decisions dominate PED-X-Bench (60%, 445/737), *Partial* accounts for 31%, whereas *Full* borrowing is rare (3%) and 6% of records are *Unlabeled*. Outcome frequency co-varies with the legislative route: BPCA submissions are overwhelmingly *None*, whereas PREA-only programs obtain *Partial* or *Full* borrowing almost twice as often. Age coverage, rather than study size, distinguishes the classes: *Full* approvals span the entire pediatric spectrum, yet even they rarely cite more than three pediatric studies.

Study-design flags (Fig. 2) reinforce this narrative. Classical efficacy and safety trials appear in >60 % of *None* labels but in <25 % of *Full*, indicating that when extrapolation is granted, sponsors rely primarily on pharmacokinetic/pharmacodynamic bridging rather than new randomised efficacy studies. Dose-escalation and neonatal studies remain scarce across the board.

### 5.2  ABLATION ANALYSES

Table 3: Ablation results for the LLM labelling pipeline with 95% bootstrap confidence intervals.

| Variant | Accuracy (95% CI) | Macro-F1 (95% CI) | AUC |
|---|---|---|---|
| Two-stage (*o3-mini*) | 0.740 [0.680–0.795] | **0.633 [0.521–0.716]** | 0.722 |
| Two-stage (*gpt-4o-mini*) | 0.534 [0.442–0.614] | 0.294 [0.210–0.365] | 0.602 |
| Single-stage (full label) | 0.725 [0.643–0.800] | 0.637 [0.532–0.708] | 0.619 |
| Single-stage + verifier | 0.804 [0.726–0.874] | 0.538 [0.436–0.615] | 0.706 |

**Pipeline Variants**  Table 3 shows that swapping *o3-mini* for *gpt-4o-mini* in the two-stage pipeline drops accuracy from 0.74 to 0.53 and macro-F1 from 0.63 to 0.29, largely due to hallucinated rationales. A single-stage prompt recovers accuracy but lowers agreement with reviewers. Adding a verifier boosts accuracy to 0.80 but collapses many borderline cases into the majority class, lowering balanced F1. Statistical significance testing (Appendix F.3) confirms these differences. Hence we keep the two-stage *o3-mini* pipeline as the official labeller.

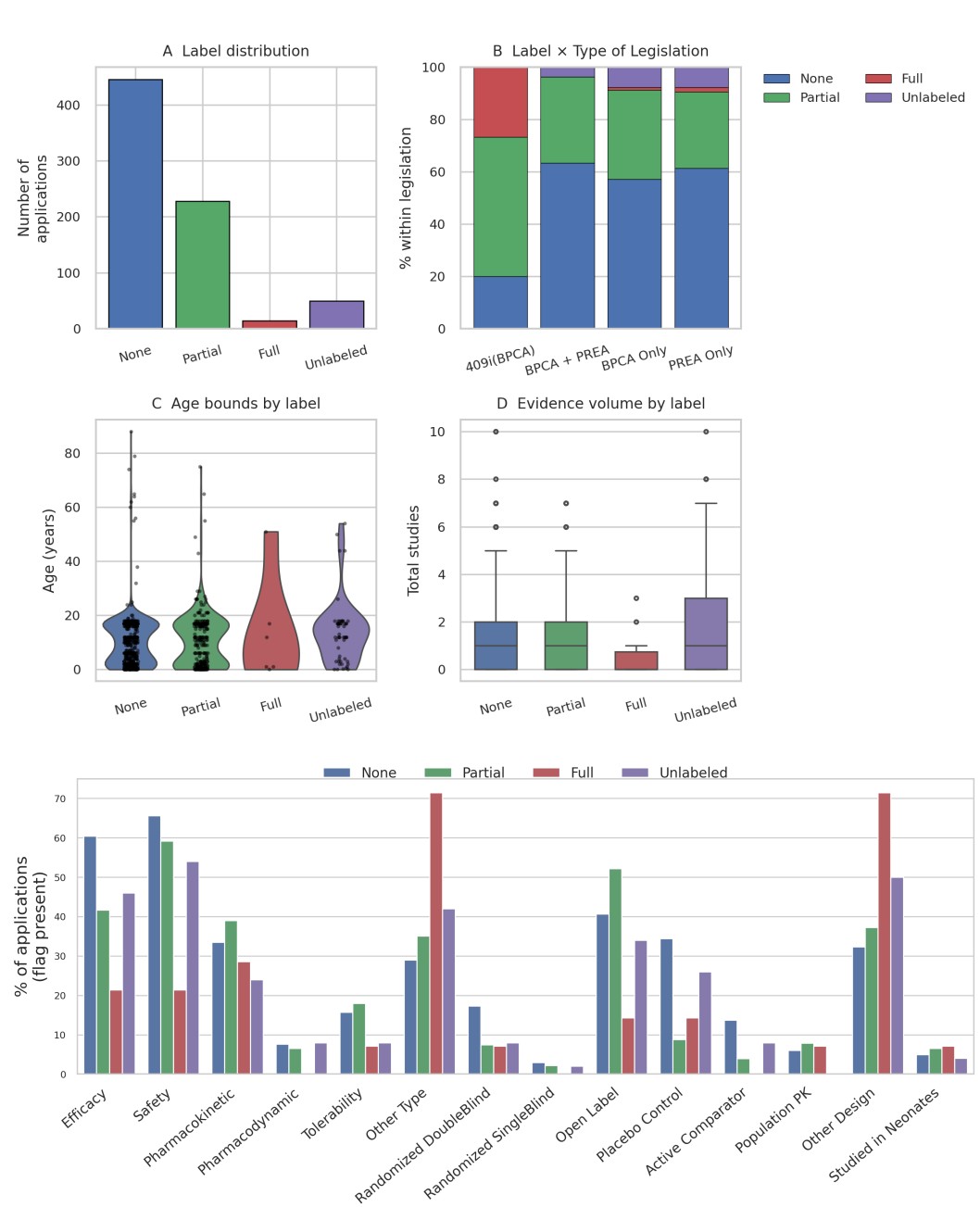

Figure 2: Dataset overview and characteristics. **Top:** Corpus statistics showing (A) label distribution across extrapolation categories, (B) extrapolation outcomes by legislative pathway (BPCA vs. PREA), (C) age-range coverage for each class (violins show min–max bounds), and (D) number of pediatric studies cited per label. **Bottom:** Study-design and outcome flags by extrapolation class, with bars denoting the percentage of applications with each characteristic.

**Qualitative Error Analysis** To better understand the differences observed in the ablation results, we conducted a qualitative analysis of labeling errors. Manual inspection of model outputs revealed two primary error types:

**Misinterpretation errors:** Models sometimes confuse study outcomes with study conduct. For example, CIMZIA was evaluated for pediatric Crohn's disease in 99 subjects aged 6–17 years, but "efficacy was not demonstrated" due to high discontinuations. The model incorrectly classified this as "Unlabeled," even though a pediatric efficacy study was performed. Our annotation guidelines emphasize whether studies were conducted, not whether they succeeded, but models often struggled with this nuance.

**Omission errors:** Models also fail to extract complete evidence, particularly pharmacokinetic and safety data buried in complex label sections. This leads to systematic underestimation of pediatric evidence and misclassification of "Partial" or "Full" cases as "None" or "Unlabeled." Missing PK/PD mentions in the summary stage often propagated into final classification errors.

These error types are reflected in the confusion matrix (Fig. 3). The model performs strongly on the majority *None* class but often confuses *Partial* with *Unlabeled*, while rare *Full* cases remain too sparse for robust assessment. These results highlight the challenge of distinguishing borderline categories and the need for class-balanced objectives. The confusion patterns reveal that most classification errors occur at the boundaries between similar extrapolation types, suggesting that the regulatory decision space contains inherent ambiguity that even expert annotators sometimes struggle to resolve consistently. We further illustrate these "borderline" cases with detailed qualitative examples in Appendix E.

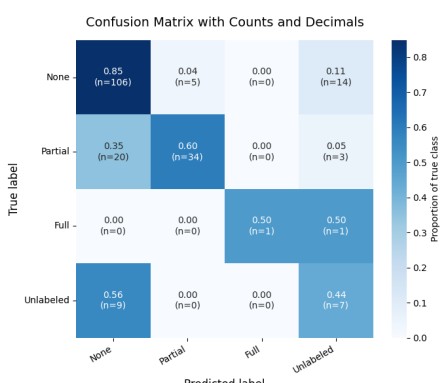

Figure 3: Confusion matrix with normalized values and counts.

**LLM pipeline model comparison** Table 4 shows meaningful variation in performance across LLM families, with no single model dominating all metrics. Gemini 2.5 Pro attains the highest accuracy (0.772), while GPT o3-mini achieves the best macro-F1 (0.633) and AUC (0.788), indicating superior handling of rare but critical categories such as *Full*. Claude Sonnet 3.7 performs consistently lower, especially on macro-F1 (0.392), suggesting challenges in balanced classification.

Table 4: LLM pipeline performance across model families on expert-annotated labels (135 samples) with 95% bootstrap confidence intervals.

| Model | Accuracy (95% CI) | Macro-F1 (95% CI) | AUC |
|---|---|---|---|
| Gemini 2.5 Pro | 0.772 [0.694–0.841] | 0.602 [0.493–0.685] | 0.755 |
| GPT o3-mini | 0.740 [0.680–0.795] | **0.633 [0.521–0.716]** | 0.788 |
| Claude Sonnet 3.7 | 0.704 [0.615–0.777] | 0.392 [0.301–0.470] | 0.617 |

These results provide confidence that our annotation pipeline's effectiveness is not an artifact of a single model architecture, while also highlighting the importance of careful model selection for specialized regulatory text classification tasks. We adopt GPT o3-mini as the official annotator due to its superior balance across metrics, particularly its performance on minority classes.

## 5.3 BASELINE CLASSIFIERS

Table 5: Baseline performance on the blind test split (34 labels) with 95% bootstrap confidence intervals.

| Model | Accuracy (95% CI) | Macro-F1 (95% CI) | AUC |
|---|---|---|---|
| Logistic regression (metadata) | 0.647 [0.56–0.73] | 0.359 [0.26–0.45] | 0.650 |
| XGBoost (metadata) | 0.676 [0.59–0.76] | 0.339 [0.24–0.43] | 0.660 |
| BigBird RoBERTa (finetuning) | 0.678 [0.60–0.77] | 0.369 [0.28–0.47] | 0.839 |
| BigBird (linear probing) | 0.610 [0.52–0.70] | 0.190 [0.12–0.28] | 0.829 |
| ClinicalModernBERT (finetuning) | 0.750 [0.66–0.84] | 0.450 [0.35–0.54] | 0.890 |
| ClinicalModernBERT (linear probing) | 0.810 [0.73–0.89] | **0.525 [0.42–0.62]** | 0.850 |
| BioclinicalBERT (finetuning) | 0.670 [0.58–0.76] | 0.300 [0.21–0.39] | 0.850 |
| BioclinicalBERT (linear probing) | 0.610 [0.52–0.70] | 0.490 [0.39–0.59] | 0.837 |
| XGB fusion (metadata + full text) | 0.765 [0.68–0.85] | 0.391 [0.30–0.49] | 0.850 |

As shown in Table 5, domain-adapted transformers substantially outperform both metadata-only models and generic architectures. ClinicalModernBERT achieves the best performance (0.81 accuracy, 0.525 macro-F1) in linear probing mode, indicating that pretrained representations already encode much of the relevant signal. Interestingly, linear probing often matches or exceeds full fine-tuning, suggesting limited gains from end-to-end training at this dataset scale. The XGB fusion model (0.765 accuracy, 0.391 macro-F1) improves on metadata-only baselines but remains below text-only models, underscoring that most signal resides in the label text. Although logistic regression and XGBoost perform competitively using only coarse descriptors, domain-specific models deliver a clear +0.16 accuracy gain. Together, these results validate PED-X-Bench as a challenging benchmark and reveal significant headroom beyond current methods (best macro-F1 = 0.525). Detailed descriptions of baseline architectures, training configurations, and implementation specifics are provided in Appendix F.4.

## 6 DISCUSSION

We presented **PED-X-Bench**, the first large-scale benchmark that encodes FDA pediatric extrapolation decisions as a four-way classification task, linking each outcome to its supporting rationale and study metadata. Our two-stage LLM pipeline achieves substantial agreement with domain experts ($\kappa = 0.79$) and establishes strong baselines across metadata-based, transformer-based, and fusion models. These results demonstrate that regulatory reasoning tasks of this complexity are tractable, yet far from solved.

PED-X-Bench transforms over two decades of FDA regulatory decision-making into a structured, machine-readable resource, enabling analyses and applications that were previously infeasible. The benchmark captures well-known but poorly quantified trends—such as the rarity of full extrapolation, the prevalence of partial extrapolation in PREA submissions, and persistent evidence gaps in neonates—with far greater granularity than earlier manual reviews. By connecting categorical outcomes to textual rationales, it turns a historically opaque decision process into a tractable learning problem that can support the next generation of decision-support tools.

The broader implications of this work extend well beyond benchmark construction. Predictive use of PED-X-Bench could accelerate pediatric drug development by identifying cases where dedicated efficacy trials are unnecessary, saving an estimated $10–15 million per avoided study and bringing pediatric labeling to market two to three years faster Li et al. (2007). Systematically codifying extrapolation precedents can also help reduce off-label prescribing which currently accounts for over 40% of pediatric use and up to 90% in neonates — by supporting tools that convert such prescribing into safer, evidence-based, on-label practice. Furthermore, PED-X-Bench introduces a new level of regulatory transparency: by surfacing over 100,000 pages of FDA text in a structured, searchable format, it enables precedent retrieval, consistency audits, and more strategic targeting of pediatric research funding toward populations where extrapolation repeatedly fails. Beyond these applications, PED-X-Bench could also serve as the foundation for decision-support systems that

assist regulators and sponsors in real time, providing evidence-grounded recommendations on when adult data are likely sufficient and when dedicated pediatric studies remain essential.

**Limitations and Future Work**   Despite its contributions, this work has several limitations. First, the dataset inherits the biases of publicly available labels: negative trials, interim analyses, and unpublished regulatory correspondence are absent, so PED-X-Bench necessarily reflects the final, public-facing narrative rather than the full deliberative record. Second, class imbalance is severe, with only 22 fully extrapolated cases; while this mirrors real-world rarity it also constrains model learning. Third, extrapolation decisions evolve as new pharmacology or real-world safety data emerge; our annotations capture a single snapshot (2007-2024) and will require periodic refreshes to stay current. And finally, the current annotations address extrapolation at the level of the entire product rather than at the granularity of individual indications or dosage forms, a simplification that future work could refine.

Looking forward, PED-X-Bench can serve as the foundation for more advanced decision-support systems that integrate external evidence from clinical trial registries, population-PK models, disease-similarity networks, and real-world safety data. Retrieval-augmented approaches, in particular, could contextualize label text with historical precedents, while multi-modal architectures could reason across pharmacologic, clinical, and regulatory signals. Such advances would not only enhance predictive accuracy but also support more transparent, evidence-grounded decision-making — ultimately accelerating pediatric drug development and improving therapeutic safety for children.

## 7   CONCLUSION

PED-X-Bench transforms two decades of FDA labeling into the first open, machine-readable resource for studying pediatric extrapolation, coupling four-way outcome labels to their precise textual rationales and rich study metadata. Through a comprehensive suite of benchmarks, we show that this task is both tractable and far from solved. By providing a rigorously curated corpus, baseline scores, and an extensible LLM annotation pipeline, we lay the groundwork for future systems that can reason across trials, pharmacology, and real-world evidence to deliver transparent, data-driven extrapolation recommendations and thereby, advancing both clinical NLP research and pediatric drug development.

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

# A  APPENDIX

## SUPPLEMENTARY MATERIAL

# B  DATASET CONSTRUCTION AND VALIDATION

## B.1  SOURCE DATA AND MOTIVATION

Pediatric labeling submitted to the U.S. Food and Drug Administration (FDA) provides the most authoritative record of how clinical evidence informs pediatric drug approvals. These updates often include new dosing, safety, or efficacy information, sometimes representing entirely new pediatric indications, while others introduce dosing restrictions or new safety warnings. Such changes are the primary mechanism by which pediatric clinical evidence informs treatment decisions.

Since the introduction of the Best Pharmaceuticals for Children Act (BPCA, 2002), the Pediatric Research Equity Act (PREA, 2003), and the Pediatric Rule (1998), the FDA has released two complementary public datasets summarizing the outcomes of these legislative efforts:

- **Pediatric Labeling Changes Spreadsheet:** A historical summary of all pediatric labeling updates since 1998, including product name, indication, legislation type, and a textual summary.

- **Pediatric Study Characteristics Spreadsheet:** A structured dataset describing the clinical studies that supported each labeling change, including study design, sample size, age range, study centers, and geographic scope.

For **PED-X-Bench**, we built upon the *Pediatric Study Characteristics* dataset because it contains the granular, structured metadata necessary for machine learning applications. These fields enable richer feature extraction, metadata-based classification, and model interpretability compared to the higher-level labeling-change spreadsheet.

## B.2  DOCUMENT ACQUISITION AND PROCESSING

Each entry in the FDA dataset is associated with a canonical submission identifier (`canon_id`, e.g., `NDA_050441_0086`). Using these identifiers, we programmatically retrieved the corresponding FDA product labeling PDFs from their public links. Our custom download pipeline implemented content-type verification, retry logic, and user-agent spoofing to ensure reliable retrieval.

PDFs were then converted into plain-text files using `pdfminer.six`, preserving structure where possible and handling encoding errors gracefully. This process yielded a parallel corpus of structured metadata and unstructured regulatory narratives suitable for downstream NLP analysis.

## B.3  GOLD-STANDARD SUBSET AND EXPERT ANNOTATION

To support robust validation and benchmarking, we curated a gold-standard subset of 135 drug labels. Sampling was stratified across diverse therapeutic domains to capture a broad range of regulatory reasoning scenarios (Table 6).

Nine domain experts (eight biomedical data scientists and one clinician) independently annotated each label following a standardized protocol. Annotators recorded: (1) the extrapolation outcome (`Full`, `Partial`, `None`, or `Unlabeled`); (2) verbatim evidence excerpts with page references; and (3) notes on ambiguity or uncertainty. Each annotation required $\sim$45 minutes due to the complexity of long-form regulatory documents. Disagreements were resolved through consensus review, yielding a gold-standard set with substantial agreement ($\kappa = 0.79$).

## B.4  VALIDATION AND DISAGREEMENT ANALYSIS

Inter-annotator agreement varied by category, with highest concordance for `None` ($\kappa = 0.804$) and `Partial` ($\kappa = 0.806$), and lower scores for `Unlabeled` ($\kappa = 0.757$) and `Full` ($\kappa =$

−0.006) due to rarity and intrinsic ambiguity. Approximately 21% of cases showed disagreement, highlighting the nuanced and context-dependent nature of pediatric extrapolation decisions.

These disagreement patterns align closely with model errors (see Section E) and underscore the inherent interpretive complexity of extrapolation decisions — particularly at category boundaries where even domain experts diverge.

Table 6: Therapeutic area distribution of the 135 manually annotated labels used for expert adjudication.

| Therapeutic Area | Count |
|---|---|
| Oncology Support / Biologic | 32 |
| Gastrointestinal & Motility | 28 |
| Antivirals | 24 |
| Respiratory (Asthma / RSV) | 20 |
| Endocrine / Metabolic | 18 |
| Central Nervous System / Psychiatry | 13 |

## B.5 DATASET

### B.5.1 DATASET STRUCTURE

**Label Distribution.** PED-X-Bench contains 737 labeled drug applications annotated into four extrapolation categories (Table 7). The majority of submissions have sufficient pediatric data to support independent approval, while *Partial* cases rely on a mix of adult efficacy and pediatric PK/safety bridging. *Full* cases remain rare, reflecting the limited situations in which adult evidence alone is considered adequate.

Table 7: Label distribution in PED-X-Bench.

| Label | Count | % | Definition |
|---|---|---|---|
| None | 445 | 60.4% | Independent pediatric evidence; no extrapolation required |
| Partial | 228 | 30.9% | Mixed evidence; partial extrapolation from adult data |
| Unlabeled | 50 | 6.8% | Insufficient information for classification |
| Full | 14 | 1.9% | Adult data considered sufficient for pediatric use |

**Study Types.** Most submissions are supported by randomized controlled trials (RCTs), often complemented by pharmacokinetic and safety bridging studies (Table 8). The relatively small number of PK-only submissions reflects the limited circumstances in which exposure-matching alone is sufficient for regulatory approval.

Table 8: Study type distribution in PED-X-Bench.

| Study Type | Count | % |
|---|---|---|
| Randomized Controlled Trials (RCT) | 459 | 62.3% |
| PK + Safety Studies | 211 | 28.6% |
| PK-Only Studies | 14 | 1.9% |
| Other / Unspecified | 53 | 7.2% |

**Data Fields.** Each PED-X-Bench entry links three complementary data layers that together enable structured modeling of regulatory decisions:

- **Classification Fields:** Canonical FDA identifier (`canon_id`), extrapolation label, gold-standard indicator, and expert confidence level.
- **Evidence Fields:** Summaries of efficacy results, pharmacokinetic and safety findings, and rationale for the assigned extrapolation outcome.
- **Metadata Fields:** Drug name, indication, therapeutic category, approval date, pediatric age range, sample size, and study design characteristics (e.g., randomization, blinding, control type).

Together, these structured fields and textual rationales enable both supervised learning tasks (e.g., classification and retrieval) and exploratory analyses (e.g., evidence mapping, multi-modal modeling, and explainable NLP).

## C ANNOTATION GUIDELINES

This section reproduces the detailed annotation protocol that was shared with domain experts prior to manual review. The guidelines were designed to ensure consistency in interpreting pediatric labeling language, standardize evidence extraction, and align all annotators on the definitions and decision rules used throughout the project. They formed the basis of the gold-standard adjudication process described in Section D

### C.1 EXTRAPOLATION TAXONOMY

Our four-way classification system captures the spectrum of pediatric extrapolation decisions based on the evidence patterns present in FDA drug labels.

Table 9: Extrapolation category definitions with evidence patterns and typical phrases.

| Category | Evidence Pattern Required | Conceptual Framework | Typical Phrases |
|---|---|---|---|
| **None** | Pediatric efficacy RCT or adequate & well-controlled study per 21 CFR 314.126 measuring clinical endpoints. Adult trials may be cited but pediatric efficacy is proven independently. | No extrapolation needed | "randomized, double-blind, placebo-controlled trial in patients 1 month–16 years"; "adequate and well-controlled pediatric study met primary endpoint" |
| **Partial** | No stand-alone pediatric efficacy RCT. Label includes pediatric PK, exposure–response modeling, or safety-only cohorts. Efficacy inferred from adult RCTs with pediatric bridging data. | Adult RCTs provide efficacy evidence; pediatric data bridges dose/exposure or confirms safety | "population pharmacokinetic study in 24 pediatric patients"; "open-label safety cohort"; "dosing selected to match adult AUC" |
| **Full** | No pediatric trials. Label provides only PK simulations, allometric scaling, or weight-based dosing tables. Both efficacy and safety extrapolated from adults. | PK-only bridge | "Dose in children derived by allometric scaling"; "Exposures in pediatric patients expected to match adults" |
| **Unlabeled** | Label states "Safety and effectiveness in pediatric patients have not been established" with no PK or dosing information provided. | No pediatric information | Exact phrase or very similar with no other pediatric sections |

### C.2 ANNOTATION PROTOCOL

Annotators followed a systematic five-step process for each drug label:

**Step 1: Label Access**    Access FDA drug labels via provided URLs or search FDA's Drugs@FDA database using application numbers for broken links.

**Step 2: Section Identification**    Use text search ("pediatric") to locate relevant sections, typically found in:

- Adverse Reactions section
- Section 8.4 – Pediatric Use
- Clinical Studies sections (§14.x)
- Clinical Pharmacology §12.3 (for PK data)

**Step 3: Evidence Extraction**    Extract verbatim text (1–3 sentences, ≤600 characters) documenting:

- **Efficacy evidence:** Phrases indicating randomized, controlled trials or adequate studies with clinical endpoints
- **PK/dosing evidence:** Pharmacokinetic studies, weight-based dosing tables, exposure-matching statements, or simulation results

**Step 4: Classification**    Assign study type (RCT, PK+Safety, PK Only, or custom descriptor) and extrapolation category using the taxonomy above.

**Step 5: Quality Control**    Cross-reference extracted information with metadata fields (e.g., "Ages Studied") and flag ambiguous cases for expert adjudication.

### C.3    EDGE CASES AND DECISION RULES

**Age boundaries:**    Participants ≥17 years treated as adults; pediatric-specific labeling ignored.

**Mixed study designs:**

- Pediatric arm within adult efficacy RCT: *None* if randomized with efficacy analysis; *Partial* if safety/PK only
- Safety-only randomized studies: *Partial* (randomization does not imply efficacy assessment)
- Open-label safety cohorts: *Partial* (never *Full* when pediatric data exist)

**Multiple indications:**    If any pediatric indication relies on adult efficacy plus pediatric bridging data, classify entire label as *Partial*.

All annotators received training on these guidelines and completed practice annotations before independent review of assigned drug labels.

## D    INTER-ANNOTATOR AGREEMENT ANALYSIS

Table 10 summarizes representative cases of disagreement between expert annotators, illustrating key sources of ambiguity and boundary challenges in pediatric extrapolation decisions.

## E    EXPERT–LLM DISAGREEMENT CASE STUDIES

Table 11 presents representative examples of disagreement between expert annotations and LLM predictions, highlighting key failure modes and their clinical implications.

Table 10: Representative sources of expert disagreement, illustrating the nature of regulatory ambiguity and boundary challenges.

| Disagreement Type | Example | Summary and Implications |
|---|---|---|
| **Pediatric Studies vs. Adult Extrapolation** (None → Partial) | *Zinplava (bezlotoxumab)* | Experts disagreed on whether language such as "supported by evidence from adult trials with additional pediatric PK and safety data" constituted extrapolation. One annotator interpreted the pediatric data as independent establishment of efficacy (None), while another viewed the adult evidence as central to approval (Partial). |
| **Ambiguity in Evidence Completeness** (Unlabeled ↔ None / Partial) | Multiple cases | Disagreement often arose when limited pediatric data existed (e.g., PK bridging or safety studies) but were insufficient for efficacy demonstration. Annotators differed on whether to label such cases as Unlabeled (insufficient evidence) or Partial (some evidence bridging). |
| **Extreme Rarity and Definition Uncertainty** (Full) | Rare instances | The single "Full" case showed total disagreement due to insufficient precedent and lack of clear criteria for complete extrapolation. Experts disagreed on whether PK and safety data alone were sufficient to support efficacy without pediatric efficacy data. |

Table 11: Representative disagreement cases between expert adjudication and LLM predictions, illustrating key failure modes and clinical implications.

| Case | True Label | LLM Prediction | Summary and Implications |
|---|---|---|---|
| **1. Under-classification of Evidence Requirements** *Ramucirumab (Cyramza)* | Partial | None | LLM misclassified a case with pediatric PK and safety data (N=23) as *None*, failing to account for the need for randomized efficacy data in oncology and additional evidence due to juvenile growth plate toxicity signals. |
| **2. Over-conservative Evidence Assessment** *Dupilumab (Dupixent)* | None | Unlabeled | Robust pediatric evidence from a Phase 3 RCT (N=408, ages 6–11) demonstrated significant efficacy and safety comparable to adults. The LLM misinterpreted complexity from multi-indication approvals and age-specific dosing adjustments as evidence insufficiency. |
| **3. Borderline Evidence Complexity** *Bedaquiline* | Unlabeled | None | Compassionate use data showed culture conversion but lacked controlled trials and raised QT prolongation concerns. Experts deemed the evidence insufficient (*Unlabeled*), while the LLM over-interpreted surrogate outcomes. |

# F LLM PIPELINE DETAILS

## F.1 MODEL CONFIGURATION AND HYPERPARAMETERS

We evaluated three large language model (LLM) families using identical two-stage prompting pipelines with structured function or tool calling to ensure consistent JSON output formatting.

**GPT o3-mini (Azure OpenAI)**

- **API Version:** 2024-02-15-preview
- **Max Completion Tokens:** 12,000 per call

- **Function Calling:** Enforced via `function_call` parameter

- **Inter-request Delay:** 0.3 seconds (`time.sleep(0.3)`)

**GPT-4o-mini (Azure OpenAI)**

- **API Version:** 2024-02-15-preview

- **Max Completion Tokens:** 12,000 per call

- **Function Calling:** Enforced via `function_call` parameter

- **Inter-request Delay:** 0.3 seconds (`time.sleep(0.3)`)

**Gemini 2.5 Pro (Google AI)**

- **Temperature:** 0.1, **Top-p:** 0.95, **Top-k:** 64

- **Max Output Tokens:** 8,192

- **Safety Settings:** All categories set to `BLOCK_NONE`

- **Text Truncation:** 150,000 characters with intelligent break points

- **Inter-request Delay:** 2.0 seconds (configurable via `--sleep`)

**Claude Sonnet 3.7 (Anthropic)**

- **Model:** `claude-3-7-sonnet-20241022`

- **Max Tokens:** 4,096

- **Tool Choice:** Forced tool selection for structured output

- **Rate Limit Handling:** Exponential backoff (60s, 120s, 180s) with streaming fallback

- **Text Truncation:** 50,000 characters with sentence-boundary preservation

- **Inter-request Delay:** 5.0 seconds (configurable via `--sleep`)

All models processed text files with UTF-8 encoding and error tolerance. JSON serialization used compact formatting (`separators=(',',':')`) to minimize token consumption.

## F.2 PROMPT ENGINEERING

---

**Step 1: Extract from FDA Label**

**Prompt:**
You are scanning an FDA product label. Return JSON ONLY via the function call. Follow this schema exactly.
- Summarize ≤150 words.
- DO NOT invent data.

**Variables Extracted:**
- PediatricSummary
  - section (string)
  - summary (string)
- AllAges (array of strings)
- Comments (string)

---

**Step 2: Classify Extrapolation Type**

**Prompt:**
You are an expert in FDA pediatric extrapolation. Use the decision tree:
- None – ≥1 pediatric efficacy RCT
- Partial – pediatric PK and/or safety evidence but NO efficacy RCT
- Full – only PK / exposure modelling; no pediatric safety cohort
- Unlabeled – no pediatric evidence.

**Variables Extracted:**
- resolved_label (string: "None", "Partial", "Full", "Unlabeled")
- peds_study_type (string: "RCT", "PK+Safety", "PK Only", "None")
- efficacy_summary (string)
- pk_summary (string)
- lowest_age_band (string)
- highest_age_band (string)
- rationale (string)
- confidence (string: "high", "medium", "low")

---

Figure 4: Summary and classification prompts used in the two-stage pipeline.

## F.3 STATISTICAL SIGNIFICANCE TESTING

Table 12: McNemar's exact test for pairwise comparisons of pipeline variants.

| Comparison | $p$-value | Significant ($p < 0.05$) |
|---|---|---|
| Two-stage (*o3-mini*) vs Single-stage | 0.013 | Yes |
| Two-stage (*o3-mini*) vs GPT-4o-mini | 0.002 | Yes |
| Two-stage (*o3-mini*) vs Single-stage + Verifier | 0.072 | No |
| Single-stage vs GPT-4o-mini | 0.241 | No |
| Single-stage + Verifier vs GPT-4o-mini | 0.031 | Yes |

McNemar's exact test confirms that the two-stage *o3-mini* pipeline significantly outperforms both single-stage and *gpt-4o-mini* variants, while differences with the verifier-augmented model are not statistically significant.

## F.4 BASELINE MODEL IMPLEMENTATION DETAILS

**Text-Based Models** All transformer models utilized Hugging Face's `transformers` library with consistent preprocessing: UTF-8 text encoding with error tolerance, early stopping

(patience=2-4), and AdamW optimization. Models differed in tokenization limits and training paradigms:

**BigBird RoBERTa** (`google/bigbird-roberta-base`) processed 4,096 tokens with gradient checkpointing, batch size 1, and 8-step gradient accumulation. Fine-tuning used 2e-5 learning rate for 4 epochs with FP16 precision.

**ClinicalModernBERT** (`Simonlee711/Clinical_ModernBERT`) handled 8,192 tokens with two training modes: (1) full fine-tuning with 2e-5 learning rate and batch size 1, and (2) linear probing with frozen encoder and 1e-3 learning rate for the classification head only, using larger batch sizes (4) due to reduced memory requirements.

**BioClinicalBERT** (`emilyalsentzer/Bio_ClinicalBERT`) was limited to 512 tokens due to memory constraints, trained with batch size 4 and 2-step gradient accumulation. Both fine-tuning (2e-5 LR) and linear probing (scikit-learn LogisticRegression with L2 penalty, C=1.0) variants were implemented.

**Fusion Model**   The XGBoost fusion baseline combined ClinicalModernBERT [CLS] embeddings (768-dimensional) with structured metadata features. Metadata processing included: (1) standardized numeric features (patient counts, ages, study statistics), (2) binary flags for study characteristics (randomization, blinding, design type), and (3) one-hot encoded categorical variables (legislation type, therapeutic category, administration routes). The final feature vector concatenated text embeddings with 253-dimensional metadata, trained using XGBoost (max_depth=6, learning_rate=0.1, n_estimators=300) with early stopping on development loss.

**Metadata-Only Baselines**   Logistic regression employed multinomial classification with class-balanced weights, tuning penalty type (L1/L2/elastic-net) and regularization strength (C $\in$ 0.01, 0.1, 1, 3, 10) via grid search optimizing macro-F1 on the development set. XGBoost metadata baseline used identical preprocessing with hyperparameter search over tree depth (3, 6, 10), learning rates (0.01, 0.1, 0.3), and estimator counts (100, 300, 500).

**Computational Requirements**   Training times varied significantly: metadata-only models completed in under 60 seconds on CPU, while transformer fine-tuning required 2-4 hours on single A100 GPUs. Linear probing reduced training time by 60-70% compared to full fine-tuning while often achieving comparable performance, suggesting that pretrained biomedical representations already captured much of the regulatory reasoning signal.

All models used consistent train/dev/test splits with identical preprocessing pipelines to ensure fair comparison. Random seeds were fixed (seed=42) for reproducibility across all experiments.

# G   ADDITIONAL RESULTS AND ANALYSIS

## G.1   PER-CLASS PERFORMANCE ANALYSIS

Table 13: Per-class performance of the best LLM pipeline (two-stage *o3-mini*) on the expert-annotated test set, with 95% bootstrap confidence intervals.

| Class | F1 (95% CI) | Accuracy / Recall (95% CI) |
|---|---|---|
| None | 0.815 [0.770–0.857] | 0.824 [0.752–0.888] |
| Partial | 0.708 [0.600–0.808] | 0.614 [0.491–0.737] |
| Full | 0.667 [0.000–1.000] | 1.000 [1.000–1.000] |
| Unlabeled | 0.341 [0.167–0.512] | 0.312 [0.125–0.562] |

Per-class results reveal the model's performance dynamics across extrapolation decisions. The classifier achieves high precision and recall on the dominant *None* class and reasonable performance on *Partial* labels, but struggles with rare categories.

## G.2 TEMPORAL TRENDS ANALYSIS

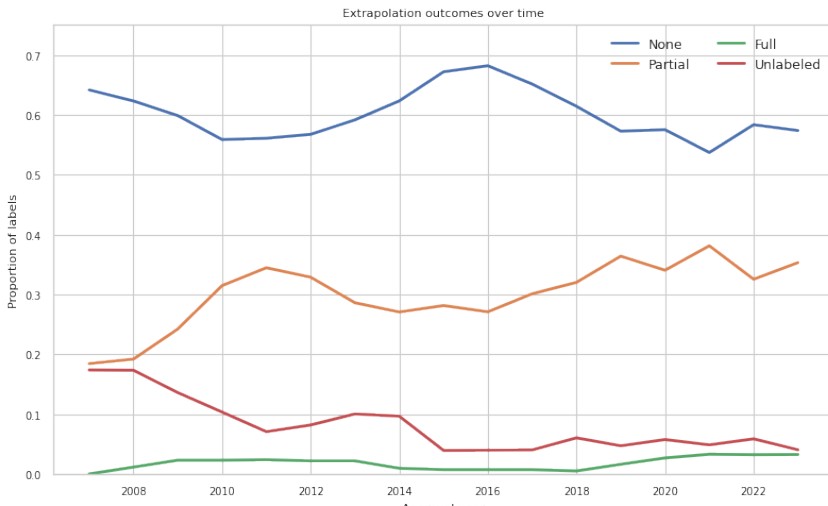

Figure 5: Trends in extrapolation outcomes over time (1998–2025). Vertical dashed line marks the adoption of ICH E11A (2024).

