# OpenReview forum: "PED-X-Bench: A Benchmark of Adult-to-Pediatric Extrapolation Decisions in FDA Drug Labels"
_ICLR.cc/2026/Conference — ICLR 2026 Conference Withdrawn Submission_

### Official Review · Reviewer_hZ4z · 2025-10-31

**Soundness:** 1
**Presentation:** 1
**Contribution:** 1
**Rating:** 0
**Confidence:** 4

**Summary:**

The paper introduces a benchmark dataset, PED-X-Bench, that captures adult-to-pediatric extrapolation decisions from the FDA. Each instance in the dataset corresponds to a regulatory decision and contains (1) the FDA drug labelling texts, (2) a categorical label on the final FDA extrapolation decision, (3) a summary of the pediatric efficacy and (4) meta-data. The stated goal is to design a benchmark dataset that can be used to design ML models to predict and analyze these decisions from the text, thereby enabling the development of decision support tools in regulatory contexts.

**Strengths:**

- The problem domain of adult-to-pediatric extrapolation is interesting and societally relevant. It would for instance be interesting to study if (and under which assumptions) the outcome of a pediatric trial can actually be modeled based on the results of a trial on adults.

- The paper makes an attempt to structure and categorize FDA regulatory decisions which are typically complex and difficult to analyze systematically.

**Weaknesses:**

The paper introduces a benchmark dataset on FDA decision-making regarding adult-to-pediatric extrapolation. While this is novel, the motivation for the dataset is lacking and no new methods are presented, making the contributions limited.
The introduction clearly describes the problem of adult-to-pediatric extrapolation, but it fails to describe the regulatory process that deals with this problem (FDA decisions pipeline, structure of the drug labels, ...). This leaves the reader guessing on the context and makes it hard to understand what is actually being modelled.
There is also no compelling motivation on why modelling FDA extrapolation decisions would be meaningful. In the conclusion it is mentioned that this can reduce the need for costly pediatric efficacy trials, but the models are trained on texts that describe the presence or absence, and the results of such trials.
Additionally, the related work section is very brief and misses several key lines of work. For instance, there are no references of LLM-assisted labelling or benchmarking with generated labels / annotations.

In summary, I identified the following weaknesses:
- (1) the paper proposes labels that are conceptually ambiguous,
- (2) these labels are assigned based on prompting LLMs instead of clear criteria,
- (3) the expert validation of the label quality is weak,
- (4) the inclusion and exclusion criteria of the paper are unclear and hard to judge, and
- (5) additional motivation and a review of existing work is also needed.

**Questions:**

- If the goal is to reduce the amount of pediatric trials, wouldn’t it be more meaningful to predict the outcome of such trails instead of the FDA extrapolation decisions?
- Would it be possible to translate the continuum of extrapolation decisions to label categories based on clear, measurable criteria?
- Since LLMs are used to assign the labels, and similar transformer-based models are used to predict these labels, how do you avoid circularity (and hence, bias) in the evaluation?

---

### Official Review · Reviewer_5sEZ · 2025-11-01

**Soundness:** 2
**Presentation:** 2
**Contribution:** 1
**Rating:** 2
**Confidence:** 2

**Summary:**

This paper presents PED-X-Bench, a benchmark dataset which contains 737 drug-labels of FDA adult-to-pediatric extrapolation outcomes as a four-way classification task. The authors employ a two-stage LLM labeling pipeline (with human expert review) and perform benchmark evaluations using this dataset.

**Strengths:**

1. Overall this paper is clearly written and easy to understand.

2. The dataset is relatively novel and generated by both LLM labeling and expert review.

**Weaknesses:**

1. The dataset size of 737 drug-labels is still small, since there are many more clinical trials conducted for pediatric conditions from 2007 to 2024.

2. Although the dataset is relatively novel, it may not be of significant interest to researchers outside this area due to its small data size, scarce positive labels, and specialized focus on adult-to-pediatric extrapolation by the US FDA.

3. The authors did not introduce novel techniques in their data curation process.

4. Related to above, the authors should clearly demonstrate that this benchmark dataset contains a representative sample of clinical trials and discuss any potential biases in the data curation process.

**Questions:**

1. What new techniques are introduced in the data curation process that are both novel and applicable to similar datasets?

2. Does this dataset contain a representative sample of adult-to-pediatric extrapolations?

3. Is the dataset quality comparable or better than manual curation by human experts in this domain?

---

### Official Review · Reviewer_AU25 · 2025-11-01

**Soundness:** 2
**Presentation:** 2
**Contribution:** 2
**Rating:** 4
**Confidence:** 1

**Summary:**

This paper introduces PED-X-Bench, the first public, machine-readable corpus that systematizes how the U.S. FDA has decided, over 2007–2024, whether adult efficacy/safety/PK data can be extrapolated to children. The authors collect 737 pediatric labeling sections from FDA pediatric labeling changes and convert them into a four-way classification task: Full, Partial, None, and Unlabeled. They do this with a two-stage LLM pipeline (LLM to extract evidence → LLM to assign the extrapolation category + rationale) and validate a stratified subset of 135 examples with nine domain experts to establish that LLM-assisted labeling is close to human reliability (κ≈0.72–0.79). They then benchmark a range of baselines from metadata-only models to domain clinical transformers, showing that current models only reach moderate macro-F1, so there is real headroom. The stated goal is to enable AI-assisted regulatory decision support and to make FDA’s historically opaque pediatric extrapolation reasoning computationally analyzable.

**Strengths:**

- Timely and non-trivial problem framing. Pediatric trials are expensive, slow, and ethically constrained; in practice regulators do extrapolate adult data, but this has been hard to study at scale because decisions are buried in PDF labels. Turning this into a structured prediction task is, in itself, a contribution.

- First public, structuring of FDA pediatric extrapolation. To my knowledge, there isn’t an open dataset that (i) spans 2007–2024, (ii) pairs each case with evidence summaries, and (iii) normalizes decisions to FDA’s own threeish buckets (full/partial/none) + “unlabeled.” That immediately creates opportunities for retrieval, RAG, policy analysis, and longitudinal studies.

**Weaknesses:**

- Goldness of the labels is partly circular and LLM-mediated.
The central object — the four-way label — is largely produced by an LLM reading FDA labels, then checked on a relatively small expert subset (135 / 737 ≈ 18%). That’s reasonable for a first release, but it also means: downstream models will be learning to mimic the LLM+guideline interpretation of FDA text, not necessarily the actual (sometimes more nuanced) FDA review logic. The paper acknowledges ambiguity (21% disagreement), but it could be clearer about which parts are unavoidably model-generated vs. truly human-confirmed. A fully human-annotated core (say 250–300 items) would make the benchmark more solid.

- Source-of-truth is the public label, not the full regulatory dossier.
FDA labels are the tip of the iceberg: nuanced reasoning, modeling assumptions (e.g., disease similarity, exposure–response modeling), and even pediatric plan negotiation often live in reviews, not labels. Training a model on labels risks learning “how FDA writes labels about extrapolation” rather than “how FDA decided to extrapolate.” The authors hint at this but don’t quantify potential information loss. This is important because the stated use case is “regulatory decision support,” which ideally wants pre-label information.

**Questions:**

N/A

---

### Note · Authors · 2025-11-14

I have read and agree with the venue's withdrawal policy on behalf of myself and my co-authors.